# Taste of Fat and Obesity: Different Hypotheses and Our Point of View

**DOI:** 10.3390/nu14030555

**Published:** 2022-01-27

**Authors:** Laurent Brondel, Didier Quilliot, Thomas Mouillot, Naim Akhtar Khan, Philip Bastable, Vincent Boggio, Corinne Leloup, Luc Pénicaud

**Affiliations:** 1Centre for Taste and Feeding Behaviour, UMR 6265 CNRS, 1324 INRAE, University of Burgundy, Franche-Comté, 21000 Dijon, France; thomas.mouillot@chu-dijon.fr (T.M.); corinne.leloup@u-bourgogne.fr (C.L.); 2Unité Multidisciplinaire de la Chirurgie de L’obésité, University Hospital Nancy-Brabois, 54500 Vandoeuvre-les-Nancy, France; quilliot.d@orange.fr; 3Department of Hepato-Gastro-Enterology, University Hospital, 21000 Dijon, France; 4Physiologie de Nutrition & Toxicologie (NUTox), UMR/UB/AgroSup 1231, University of Burgundy, Franche-Comté, 21000 Dijon, France; naim.khan@u-bourgogne.fr; 5Independent Researcher, 21121 Hauteville-lès-Dijon, France; pbastable@free.fr; 6Independent Researcher, 21000 Dijon, France; vincentboggio@wanadoo.fr; 7Institut RESTORE, Toulouse University, CNRS U-5070, EFS, ENVT, Inserm U1301 Toulouse, 31432 Toulouse, France; luc.penicaud@inserm.fr

**Keywords:** fat taste, obesity, food intake, reward system

## Abstract

Obesity results from a temporary or prolonged positive energy balance due to an alteration in the homeostatic feedback of energy balance. Food, with its discriminative and hedonic qualities, is a key element of reward-based energy intake. An alteration in the brain reward system for highly palatable energy-rich foods, comprised of fat and carbohydrates, could be one of the main factors involved in the development of obesity by increasing the attractiveness and consumption of fat-rich foods. This would induce, in turn, a decrease in the taste of fat. A better understanding of the altered reward system in obesity may open the door to a new era for the diagnosis, management and treatment of this disease.

## 1. Introduction

Recent compelling evidence has demonstrated an association between a decrease in the orosensory detection of fat and obesity. Some authors have put forward the hypothesis that the alteration in the taste of fat is simply a consequence of a high intake of fatty foods while for others, mechanisms involved in the onset of obesity are linked to a decrease in the taste of fat. The first part of this narrative review covers the importance of fat detection during food intake according to different theories, and the evolution of fat taste after bariatric surgery. Then, we highlight arguments indicating that impaired fat taste perception/sensitivity in obesity could stem from overactivation of the brain reward system, leading to an increase in the consumption of foods rich in lipids, carbohydrates and energy, and subsequently to overweight and obesity.

## 2. Teleonomy of the Taste for Fat

For a living organism, the supply of energy from carbohydrates and fats is vital. Interestingly, energy utilisation from proteins is limited except during fasting and some pathophysiological circumstances [1]. In humans, a deficit of carbohydrates or fats in the body prompts the individual to seek and ingest the missing nutrient. Conversely, an excess of either of these nutrients causes the individual to avoid them in the diet. This mechanism, leading to a satisfactory nutritional state, was described in the famous observations conducted from 1928 to 1939 by Clara Davis [2,3]: every day for at least six months, newly weaned children freely composed their meals by choosing 12 foods from a selection of 34. The children’s spontaneous choice ensured satisfactory nutritional intakes. The author concluded: “Such juggling and successful nutritional balance when more than 30 foods are presented […] suggest the existence of an innate and automatic mechanism that directs food choices, of which appetite is only one part”. Similarly, in animals, numerous studies have confirmed the reality of the glucostatic [4,5,6] and lipostatic [4,7] theories, i.e., when a nutrient is in excess or lacking in the organism, the animal avoids or seeks out the nutrient concerned. In the same way, we have observed in human subjects that when internal carbohydrate reserves are low (i.e., when the subjects are in lipid metabolism, the individuals appreciate carbohydrate-rich foods and avoid fat-rich ones [8]. The body is, therefore, able to identify the macronutrient that it needs from food, mainly through a gustatory cue [9] and to a lesser extent through olfaction [10,11,12], vision, somaesthesia, trigeminal sensations and even audition [13]. Subsequently, cognitive and environmental factors contribute to food choices [14,15,16].

To ensure a satisfactory lipid intake, the body must be able to identify lipids by their orosensory properties (discriminative component), thereby allowing the brain to recognise the physico–chemical nature of the fat-rich food in the mouth. At the same time, these stimuli, like all sensory stimuli, induce a sensation of pleasure or displeasure in the consumer (affective or hedonic component) [17]. Palatability, the hedonic component of food, is generally associated with a high-energy content (foods rich in sugar or fat) [18,19,20]. Indeed, it is food pleasure that guides the consumer towards an adapted behaviour: acceptance or rejection of food intake [15,21]. Thus, palatability reflects the usefulness of the stimulus at a given moment and triggers motivation to ingest or reject food [22]. The distinctions between different components, i.e., discriminative and hedonic components, and their functional and anatomical overlaps at the central level, have been partly described [23,24]. In keeping with these observations, in fat taste perception studies, the discriminative and hedonic components can be analysed separately, as they complement each other in guiding ingestion to ensure good nutritional and energy status.

Concerning the discriminative components, the mechanisms at both oral and intestinal levels (receptors, pathways and nerve centres) have been the subject of numerous publications [12,25,26,27] (chapter 3 in [28]). However, there is still one question that remains unanswered: whether “fat taste” is a “primary taste” or a “mouthfeel” of fat (alimentary taste) [25,29,30,31]? There are a number of elements to consider:Gustatory stimulation linked to fat is analysed by its qualitative and quantitative components thus indicating that the taste of fat/mouthfeel of fat exists [32].Fatty acids in the mouth induce gustatory evoked potentials, thus showing that fatty acids are well perceived at the cortical level, as we have observed [33].The presence of fatty acids in the mouth is perceived and processed by the central nervous system, thereby inducing adapted behaviour and anticipatory vegetative reactions [34].

Regarding the physiology of fat taste perception, it has been shown that lipid gustatory cues follow the same criteria as those for other taste qualities. Indeed, fat detection is brought about by tongue lipid receptors/sensors (CD36 and GPR120), localised in taste bud cells. The activation of these receptors by dietary fatty acids triggers an increase in the free intracellular calcium concentration, leading to transmission of the gustatory message to afferent nerves that connect, via the nucleus tractus solitarius, different areas of the brain to modulate fat eating behaviour. Clearly, the perception of the sweet taste is sharper than that of fat, which is probably linked to a number of factors (the abundance of carbohydrates in nature, lower energy density of sugars than fats, energy storage of carbohydrates in the body that is 100 times smaller than fats, the impossibility of the body to convert fats into carbohydrates and the need for a supply of glucose for the nervous system etc.). Energy storage in the form of lipids is a common feature in living organisms [the human adult energetic reserve of carbohydrates (about 350 g at the most), even if consumed in its entirety, would not cover the energy expenditure for a whole day, whereas that of lipids (about 15 kg depending on age and gender) can meet energy requirements for more than 50 days [1]; if an adult substituted his 15 kg of fat reserve for a carbohydrate reserve, he would have to carry about 60 kg (energy density and low water content in the adipose tissue). Hence, we can also cite the convincing example of mammals, which are known to stock fat to utilise it during hibernation [35].

Regarding the hedonic component of fat taste, several studies have demonstrated that animals and humans harbour a powerful attraction to fat-rich foods (Chapter 11 in [28]) [36,37]. However, it is difficult to assess what part of this attraction is specifically related to the aroma, flavour, texture and post-ingestion consequences of dietary lipids (Chapter 11 in [28]), especially since the palatability of a fat meal varies among individuals and genotypes (Chapter 16 in [28]). On the other hand, it is easy to accept that the palatability and enjoyment of foods are often related to their fat content or to the complexity of the product. For example, consumers generally prefer a slice of dry bread with butter to one without, a green salad with a vinaigrette sauce to one without, French fries or a gratin Dauphinois to boiled potatoes. It should also be noted that an intense pleasure for the taste/mouthfeel of fat seems to be acquired very early in development, since, in rodents and non-human primates, a high-fat diet in the mother during gestation and lactation, as well as exposure to a high-fat diet after weaning, are involved in programming the hedonic control of food intake in the offspring. This could be due to epigenetic changes in the promoters of specific genes within the dopaminergic reward pathways and/or the effect of metabolic hormones, such as leptin and ghrelin, on the early development of hypothalamic projections [38,39]. Note that the palatability of the fat taste is enhanced by sweetness and vice versa [28,39,40,41]. In addition, the high hedonic value of fat-rich foods is not the only cause of dysregulated body-weight normality as the high hedonic value of sweet foods also has an impact [42,43,44,45,46,47]. It has also been observed that mothers and their children who prefer a very sweet taste have a two-fold increased risk of developing obesity [48] and that those who lose the most weight after bariatric surgery are those who had the strongest attraction to sweet foods before surgery [49].

In summary, the taste and mouthfeel of fat (its texture and smoothness) identify fat in the diet, and induce a strong hedonic sensation that guides individuals to food choices to acquire energy.

## 3. High-Fat Diet and Fat Taste

The increased consumption of fats induces a negative regulation of the receptors, which means a decrease in their sensitivity (Figure 1A). It is known that a high consumption of high-fat foods is associated with a decrease in fat taste sensitivity [50,51,52,53,54,55]. Analyses of dietary patterns show that people who consume high fat and high-energy foods have an impaired ability to taste fat [34] and have a low sensitivity to oleic or linoleic acids [34,54,56,57,58]. Correlations between high fat intake and low fat taste sensitivity have also been reported [34,59,60,61,62]. This desensitisation occurred over a fairly short time for non-esterified fatty acids [54,61,62] and following exposure to a high-fat diet in a 4-week trial [63]. However, in one study, the above desensitisation was not observed [64].

Similarly, it has been observed in mice that CD36 mRNA and CD36 receptor levels decrease during the dark period and that this change is solely dependent on the presence of fat in the diet [65]. Furthermore, the incubation of human and mouse taste bud cells with linoleic acid results in the negative feedback of CD36 receptors and the positive feedback of GPR120 receptors in the membranes. Such changes would result from the consumption of a high-fat diet [66]. Conversely, experiments using a patch-clamp technique on rat taste cells showed that a decrease in fat intake induced oral hypersensitivity to fatty acids [67] and that a low-fat diet for six or eight weeks in healthy or obese subjects increased their sensitivity to the taste of fat (upregulation of receptors) [68,69].

Other mechanisms could explain the decrease in fat taste perception. Several mechanisms of sensitisation, dishabituation and adaptation involving changes in transduction, neurotransmission and central information processing, have been described [26,66,67,70,71,72,73]. This negative feedback has been linked to salivary lipase [74], salivary composition [75,76], obesity-related inflammation [77], hormonal impregnation [78], lingual or intestinal microbiota [79,80,81] and intestinal lipid metabolites [82]. Negative feedback mechanisms affecting taste sensations according to the nature of the usually consumed food have been described for salty [79,83,84,85,86], sweet [55,78,79,85,87,88], bitter [79] and umami [89], although one study did not observe this phenomenon [90]. For example, high dietary salt intake, as assessed by 24-h urinary sodium excretion, is associated with the decreased perception of salty taste (high detection thresholds) [91].

In summary, the high consumption of fat-rich foods could decrease fat taste sensitivity through a negative feedback mechanism. [note: in this paper, a parallel between perception/sensitivity of fat and obesity on the one hand, and auditory acuity and music addiction on the other hand, is presented to illustrate the issue. Therefore, by analogy, we indicate in this paragraph that regular high sound intensity can lead to a decrease in auditory perception]. Some authors have hypothesised that decreased sensitivity to the fat taste is simply a consequence of high fat intake (Figure 2).

## 4. Taste of Fat and Obesity

In 2005, Jean Pierre Montmayeur, at the Centre des Sciences du Goût et de l’Alimentation (CSGA), identified in collaboration with Philippe Besnard, the existence of CD36 as a fatty acid sensor in the gustatory papillae on the mouse tongue epithelium [92]. In their article, they stated, “These findings demonstrate that CD36 is involved in oral LCFA (long chain fatty acids) detection and raise the possibility that an alteration in the lingual fat perception may be linked to feeding dysregulation… that might influence obesity risk”. To date, more than 550 articles concerning fat taste in obesity are referenced in PubMed. Subsequently, Besnard, Khan and Mattes et al. argued this hypothesis through several studies and reviews [60,93,94]. A book on fat taste that repeatedly addresses obesity has also been published [28]. They suggest that a decrease in the perception of fatty taste could foster the attraction and consumption of fat-rich foods to induce an optimal hedonic sensation. The increased consumption of energy-rich foods would then promote the development of obesity (Figure 1B).

A decrease in fat taste sensitivity has been observed in animal models and humans. Rats and mice rendered obese by a high-fat diet are unable to detect low concentrations of fat in licking tests [95,96]. In humans, an increase in the fat (oleic acid) detection threshold is associated with an increase in body mass index (BMI) [9,50,56,57,59,60] but this association is inconsistently found [34] and even disputed [51,52,61,62,63,97,98,99,100,101], particularly in a meta-analysis [102] and a review of the literature [103]. On the other hand, some authors have observed that perceived fat taste intensity of linoleic acid [104] and the detection threshold of oleic acid are lower in overweight people [105], and that African–Americans with significant abdominal adiposity do not detect the different fat contents in salad dressings [53]. Several correlations may provide insights into the mechanisms involved in the decreased sensitivity and/or decreased perceived intensity for lipids: firstly, a decrease in the density of the fungiform papillae on the anterior part of the tongue and a negative correlation between this density and the variation in neck circumference, a marker of adiposity [106]; secondly, a mutation in the gene coding for CD36 associated with a decrease in the detection [74] or perception [58] threshold of fat (oleic and trioleic acids) in obese individuals; thirdly, a negative correlation between CD36 gene expression in obese individuals and the perception of added fat in food [107] (variants of the CD36 gene are also associated with an increase in the consumption of saturated fats in obese individuals) [108]; finally, and possibly indicating the lower sensitivity of fat receptors, studies have shown an increase in the latency of evoked potentials after stimulation by long-chain fatty acids in obese individuals as well as the absence of a reflex increase in plasma triglycerides after oral stimulation with long-chain fatty acids in obese individuals [34].

On the other hand, people suffering from obesity consume a greater proportion of high-fat foods than do normal weight people [59,109,110,111,112,113] and in environments with unlimited access to high-fat foods, the fat detection threshold of individuals is increased [19,114].

A relationship between decreased fat taste sensitivity, high-fat food consumption and obesity has thus been suggested. Some authors have hypothesised that the decrease in fat sensitivity would induce an increase in high-fat consumption to compensate for the decreased activation of dopamine D2 receptors in the motivational and reward circuits [115,116,117], which would lead to overweight and obesity.

In summary, some authors have suggested that reduced sensitivity to the taste of fat may be involved in the development of obesity (Figure 2). [Note: by analogy, a person with a hearing impairment tends to increase the sound level of his or her HIFI system in order to enjoy his or her music].

## 5. Fat Taste and Bariatric Surgery

For most authors, weight loss induced by bariatric surgery restores taste sensitivities [18,27,118]. Thus, the thresholds for detection and identification of fat are decreased (individuals become more sensitive). The consumption of fatty foods falls one month after surgery [119], and this decrease persists for 1, 6 or even 8 years [119,120,121]. Interestingly, the decrease is greater after gastric bypass than after sleeve gastrectomy [119,122,123,124,125]. It should be noted that people who have lost weight on a restrictive low-fat diet have, as after bariatric surgery, a reduced preference for fatty foods [126,127,128].

The mechanisms that may be involved in changes in fat taste perception after surgery or after restrictive diets have been reviewed in several papers [27,119,129,130,131]. Improvements in taste sensitivity (detection or identification thresholds) after weight loss are not specific to fat tastes. Indeed, although not always found [128,132,133,134], improvements in taste sensitivity have been reported for sweet tastes [124,128,129,131,135,136,137,138,139] and to a lesser degree for salty, bitter, acidic and umami tastes [97,131,134,138,139]. These improvements then lead to a homogeneous and proportional decrease in preferences and/or consumption of sweet, salty or protein-rich foods, but in a non-stereotyped and non-specific way for a given macronutrient [120,133,136,140,141,142,143,144]. It has also been reported that the greater the weight loss, the greater the improvement in taste sensitivity [125,145] and that it can sometimes even lead to aversion to certain foods, particularly when they are rich in fat and/or sugar [125,133,144,146]. Finally, it should be noted that inter-individual differences are often reported, particularly according to gender [61,124,137,147,148], and that changes in taste are not always observed [146,149]. On the other hand, it is recognised that bariatric surgery can induce improvements in olfaction as well as changes in taste [91,125,137,138,147].

In summary, impaired fat taste sensitivity is reversed by weight loss induced by bariatric surgery or calorie restriction in humans. This reversal is generally associated with a decrease in preferences for fatty foods. [Note: to continue the previous analogy, after hearing-impaired persons are fit with a hearing aid, their hearing ability improves and they reduce the volume of their HIFI systems].

## 6. Alteration of the Reward System, High-Fat Diet and Obesity

For several authors, the increase in the consumption of palatable foods rich in fat and the subsequent development of overweight and obesity is not explained by an alteration of the taste/mouthfeel of fat (qualitative and quantitative components). They highlight the importance of the hedonic component [18,95,140,144,150,151,152]: an altered reward system (causal mechanism) leads to the increased consumption of palatable high-fat foods and subsequently (i) to obesity and (ii) to the decreased taste/mouthfeel of fats via a feedback mechanism (resulting mechanism) [21,103,153,154,155,156,157,158] (Figure 1C). Numerous studies in animals have confirmed this hypothesis [21,95,159].

The comparison of the hedonic value of the taste/mouthfeel of fat by people suffering from obesity with that of normal-weight subjects is delicate, as pointed out by Linda Bartoshuk, affirming that “the hedonic properties of sweet and fat vary with body mass index: obese people live in different orosensory and orohedonic worlds than do normal-weight people. The former have lower sensitivity to sweet and fatty tastes but like sweet and fat more than the latter” [160]. Furthermore, it is difficult to reach an objective evaluation of the hedonic value of any flavour, especially in disease states [161,162,163]. Hedonic values are generally studied indirectly on the basis of eating habits, choices in experimental situations, the study of intracerebral opiates and dopamine, evoked potentials, electrical activity and brain imaging [161]. Nevertheless, as early as 1985, it was observed that the pleasure gained from consuming fatty foods was more intense in obese people than in normal-weight people [109], women preferring sweet fatty foods and men preferring salty fatty foods [164]. More recent studies have confirmed that increasing BMI was associated with increasing pleasure from the taste/mouthfeel of fat [111,160,165] and that individuals suffering from obesity report a high craving or preference for high-fat foods [19,127,166,167]. For example, the hedonic component of a graded presentation of progressively fattier foods is higher in obese individuals than in normal-weight subjects [20,168]. Similarly, a study on preferences among 10 foods with varying levels of fat showed that the preferred level correlated with the percentage of body fat [110]. It has also been reported in a prospective study over 5 years with 24,776 French people from the NutriNet-Santé cohort, that a strong preference for fat-rich foods is associated with an increased risk of obesity. In 32% of men and 52% of women, a high energy intake is explained by a strong liking for fat, whereas liking for sweetness is associated with a decreased risk of obesity [169]. Finally, it has been reported that a low-fat diet in individuals suffering from obesity restores sensitivity to fat taste/mouthfeel but does not reduce liking for fatty foods [68].

Functional magnetic resonance imaging (fMRI) studies in individuals suffering from obesity [118,136,170] have shown the overactivation of reward circuit structures during expected consumption (anticipatory food reward), involving exposure to food or food cues of palatable high-fat foods [116,171,172]. Furthermore, when normal weight subjects are shown images of food, the fMRI signal in different structures of the reward circuit (hippocampus and amygdala in the medial temporal lobe, insula, striatum, orbitofrontal cortex and ventromedial prefrontal cortex) correlates with a preference for high-fat, highly palatable foods, predicts calorie intake at the next meal [173,174] and weight gain in the following months [175,176]. Interestingly, leptin (whose plasma levels are elevated in obesity) acts on the ventral striatum to increase the palatability of food [177,178], while PYY (a satiety signal secreted by high intestinal energy and fat content and whose levels are lowered in obesity), decreases orbitofrontal cortex activation to increase food intake [179]. Positron emission tomography-scanography (PET-scan) studies have also shown that μ-opioid and dopaminergic D2 receptors are decreased in the mesolimbic system and ventral striatum of overweight and individuals suffering from obesity after food stimulation [115,117,180,181], suggesting (although this is controversial [182]) that dopamine deficiency may disrupt the eating behaviour of obese individuals [116,183,184], as observed for addictive substances in drug users [185,186,187,188]. To corroborate this hypothesis, PET-scan studies have shown that cerebral blood flow increases in the insula, part of the primary gustatory cortex, after food stimulation in people suffering from obesity [184].

Interestingly, following Roux-en-Y Gastric Bypass or sleeve gastrectomy and weight loss, patients have shown decreased activity in the mechanisms contributing to hedonic motivation for highly palatable foods, whether sweet or fatty (dopaminergic signalling and brain activity of the reward circuit), with an increase in preferences for lower calorie foods, which has a favourable influence on weight loss [49,129,130,148,189]. These changes are mainly explained by peptide YY3–36, glucagon-like peptide-1, ghrelin, neurotensin and oleoylethanolamide secretion in the ileon. However, central dopaminergic and opioid receptor signalling are the key neural mediators driving altered eating behaviour. Brain neuroimaging studies showed that brain connectivity and abnormalities are normalised following bariatric surgery [130,190,191].

A parallel can be drawn between taste and olfaction: as is the case for the taste of fat, people suffering from obesity, compared to normal-weight subjects, present lower sensitivity to food odours with a reduced ability to discriminate between them, but stronger activation of the structures of the reward circuit in fMRI. These differences are associated with the high consumption of foods rich in fat in obese individuals [174]. Finally, overactivation of the reward circuitry as well as dopaminergic alterations in response to appetitive food cues is observed in people with obesity or compulsive eating disorders. In anorexia nervosa, there is also overactivation of the reward circuit in relation to palatable food, but this is repressed by the frontal cortex so as not to allow the expression of desire and ingestive behaviour [192,193,194,195].

In summary, obesity is associated with greater brain activation of the reward circuit during the processing of food flavours and aromas, most likely due to the reinforcing value of palatable foods rich in energy, fat and carbohydrates. This could increase both energy intake and the development of obesity (Figure 2). [Note: can we consider that an overweight person who consumes food rich in sugar, fat and energy becomes a palatomane (a term derived from Old French (a palatable food is pleasant to the palate) and mania (madness, mania)), analogous to a lover of music].

## 7. Taste/Mouthfeel of Fat and Obesity, Cause or Consequence?

The data presented above show that when the food supply is varied and abundant, the reward mechanisms associated with tastes could encourage the over-consumption of palatable energy-rich foods and thereby compromise the regulation of energy balance. On the other hand, it is recognised that decreased taste/mouthfeel of fat is associated with obesity. Therefore, fat taste abnormalities may promote weight gain. However, fat taste abnormalities could also be the consequences of obesity and overactivation of the reward system.

The hypothesis that a decrease in the taste/mouthfeel of fat (causal mechanism) is partly responsible for the development of obesity runs counter to certain observations. Firstly, this association was not found in a meta-analysis [102] and is even disputed in recent studies that finely analysed fat taste sensitivity [52,62,100]. Secondly, this association seems to be dependent not on the BMI of the individuals but on their consumption of high-fat foods [52,60,62,100]. Thirdly, overactivation of the reward system in imaging studies during the presentation of high-fat foods is not specific to the taste of fat, since this phenomenon is also observed for other flavours, particularly sweetness [42,44,196,197].

Furthermore, it should be noted that hypoguesia, whether idiopathic or secondary to medical causes or induced by medication, does not lead to an increase in energy intake, but rather to its normalisation or even a decrease [198,199]. Finally, why would the reduction in the taste of fat induce overactivation of the reward system to compensate for the taste handicap rather than normal activation of the latter? In agreement with several authors [21,60,103,140,154,158], we believe that it is unlikely that obesity can be fully explained by changes in perception of the taste/mouthfeel of fat. [Note: by referring to the preceding anology, it is unlikely that one becomes a music lover because of a hearing impairment, or a film buff because of a visual impairment, or a type II diabetic because of reduced taste sensitivity for sugar].

The argument that the decrease in the taste/mouthfeel of fat in obesity stems from overactivation of the reward system (causal mechanism) is based on several facts. Firstly, the reward system directs food preferences and choices towards energy-rich foods, which therefore acquire a high hedonic value [18,37,200]. Secondly, one study reported that no change in the sensitivity threshold and no correlation with fat intake was observed in individuals suffering from obesity, while sensitivity thresholds fell in lean and overweight subjects taken together [60]. Thirdly, the same phenomenon (obesity, high food consumption and taste sensitivity reduction by negative feedback) was observed for sugar [42,160,201], and a high hedonic value for sweet foods was associated with obesity in children aged eight to fifteen years [135]. Fourth, imaging studies showed that bariatric surgery or restrictive diets decreased the reward system overactivity for palatable foods rich in fat and carbohydrates [49,129,130,189].

Furthermore, overactivation of the reward system in obesity [202,203] is associated with certain attitudes and personality traits (impulsivity, disinhibition, low self-control, etc.), hypersensitivity to certain external influences (stress, health information, etc.), dietary habits (copious meals, eating disorders etc.), the development of a sense of well-being and eating habits (heavy meals, bulimia, increased snacking, consumption of high-fat foods, etc.), all factors related to the reward system that have a major impact on eating behaviour [105,204,205,206,207,208]. These factors explain the development of obesity much better than the analysis of flavours and the taste/mouthfeel of fat. In line with the foregoing, we have observed that success in weight loss, two and a half years after a gastric bypass, is mainly linked to the psychological component of eating behaviour (emotional eating) and only to a small degree to a decrease in attraction to salty or sweet fatty foods (D. Quilliot, work in progress). It is therefore probable that overactivation of the reward system induces an increase in the hedonic components.

Therefore, overactivation of the reward system for palatable foods rich in energy could be the *primum movens* of all the abnormalities observed (energy imbalance and subsequently a reduction in the taste/mouthfeel of fat or sugar) [50,87,118]. Several questions are then raised: why is there a “pathological” overactivation of the reward circuit in obesity [209,210] and, as a result, is it conceivable that a vicious cycle is established that leads, as with hallucinogenic substances, to a real addiction to palatable energy-rich foods [200,211,212], especially as food addictions do not necessarily lead to obesity [213]? Is it a question of reduced brain control of the reward circuit by adjacent neural structures (i.e., reduced connectivity) as suggested by several recent studies on gustation and olfaction [23,116,175,214,215,216,217,218] and as observed in anorexia nervosa and eating disorders [218,219,220,221]? What influences can the indoor environment, circulating factors (triglycerides, inflammatory factors, etc.), hormones and the contents of the digestive tract or even the gut microbiota and genetic factors have on the reward system, the cognitive system and the taste/mouthfeel of fat in obesity [163,222,223,224,225,226,227]? Do endocrine disruptors (bisphenol-A) alter the reward system in the prenatal and neonatal periods as observed in mice [228]? Why do ‘pathological’ brain responses to food sometimes persist in post-obese individuals, a group at high risk of relapse [46,211,229,230]? Future studies should be able to provide some answers to these questions.

To qualify the hypotheses put forward in this presentation, it is recalled that obesity is a multifactorial disease and that obesity has numerous forms [53,163,231,232,233]. Nevertheless, this paper recommends avoiding excessive consumption of highly palatable foods in order to avoid entering a vicious cycle involving overexcitation of the reward system, as observed in addictions of various traits. This article also highlights the interest in new preventive strategies and treatment targets to help fight against energy imbalance and obesity.

In summary, the increased attraction for high-fat, high-sugar and high-energy foods, through alteration of the reward system, can at least partly explain some forms of obesity. Decreased sensitivity to dietary fatty acids may be a consequence of consumption patterns, eating behaviour and body composition.

## 8. Conclusions

The taste/mouthfeel of fat helps promote dietary fat consumption, an essential choice for calorie intake, energy storage and survival of the species. For some, the high consumption of high-fat foods by obese subjects may decrease their oral fat taste sensitivity through a negative feedback mechanism. Conversely, for others, the decrease in orosensory detection of dietary fatty acids in obese subjects may lead to an increase in fat consumption to compensate for the decrease in receptor sensitivity and in activation of D2 dopamine receptors in the motivation and reward circuits in the brain. In this paper, we hypothesised that obesity or at least some forms of obesity result from greater brain activation of the reward circuitry during the processing of food flavours and aromas, most likely due to the reinforcing value of palatable foods rich in fats, carbohydrates and energy. This hypothesis may explain, in part, both the high consumption of fatty foods and, consequently, the decrease in taste/mouthfeel of fats by negative feedback mechanisms. Furthermore, we point out that the alteration in the reward system in obesity (causal mechanism) seems to be, at least partially, reversible after weight loss induced by bariatric surgery or dieting, leading to a decrease in preferences for and consumption of fatty foods.

## Figures and Tables

**Figure 1 nutrients-14-00555-f001:**
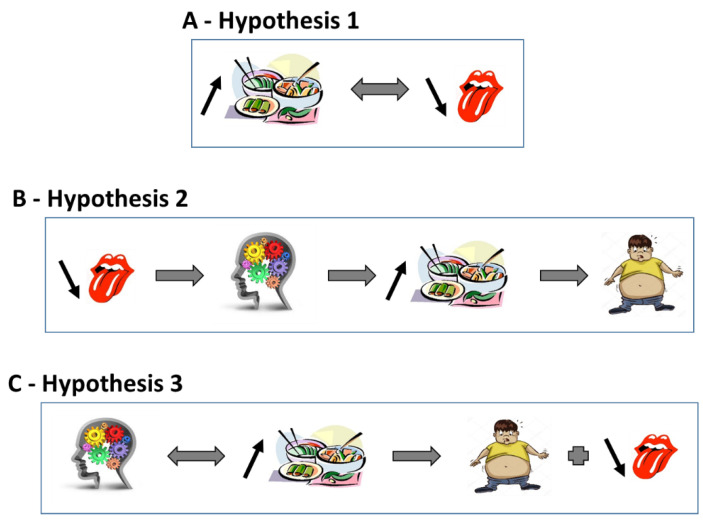
Three hypotheses have been put forward to explain the decreased taste/mouthfeel of fat in obesity. In the first (**A**), the increase in the consumption of fat-rich foods decrease the taste for fat by a negative feedback mechanism. In the second (**B**), the decrease in fat taste could increase the consumption of high-fat foods in order to activate the reward system. In the third (**C**), we assume that overactivation of the reward system could induce the high consumption of high-fat foods and consequently a decrease in fat taste. In all three situations, it is the high consumption of fat and energy rich foods that leads to overweight and obesity. ↑ and ↓ indicate respectively, increase/activate and decrease/inactivate.

**Figure 2 nutrients-14-00555-f002:**
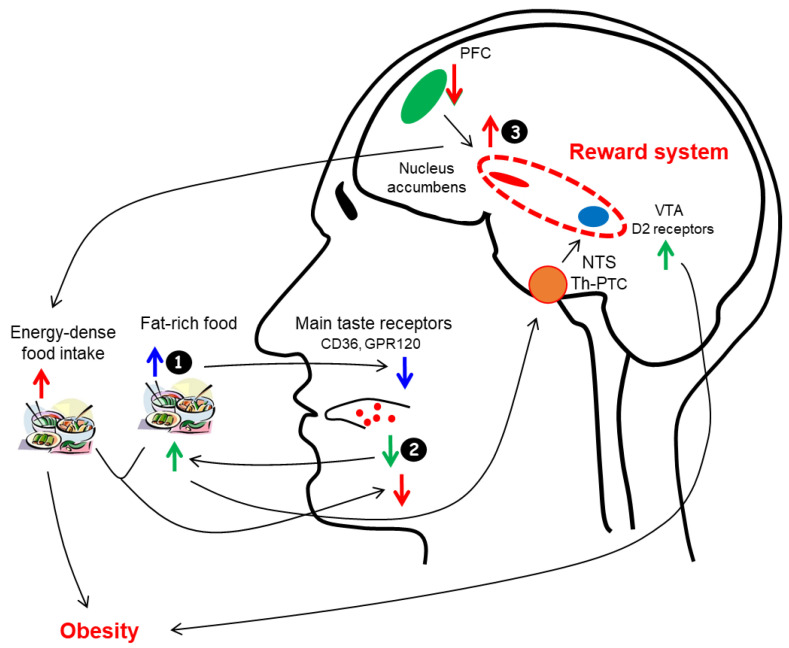
Schematic representation of “cross-talk” between fat taste perception, the brain reward system and obesity. In the first hypothesis (❶), fat-rich food intake decreases the taste for fat by a negative feedback mechanism. In a second hypothesis (❷), low fat taste sensitivity could increase the consumption of high-fat foods in order to activate the reward system. In the third hypothesis (❸), overactivation of the reward system could induce high consumption of high-energy and high-fat foods and consequently decrease fat taste sensitivity, thereby promoting the development of obesity. PFC, prefrontal cortex; VTA, ventral tegmental area; NTS, nucleus tractus solitaries; Th, ventroposteromedial nucleus of the thalamus; PTC, primary taste cortex. ↑ and ↓ indicate respectively, increase/activate and decrease/inactivate.

## Data Availability

Not applicable.

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
