# Peer review of "Taste of Fat and Obesity: Different Hypotheses and Our Point of View"

_nutrients, 2022, doi:10.3390/nu14030555_

Round 1

Reviewer 1 Report

The manuscript “Taste of fat and obesity: different hypotheses and our point of view” is a review about the relationship between fat perception and preference and obesity development. The authors, besides presenting a review of the studies where fat perception (fat taste, fat preference, etc…) was studied in the context of high-energy food intake and obesity, also presented their point-of-view about how fat perception can be linked with high-fat food preferences and obesity development.

The manuscript is interesting and contains several information.

However, it needs to improve in the way themes are organized, since, as it is, I have the idea that the authors and going back and forward trough the same concepts. There are several parts where some issues appear to be repeated and I think this is because of the organization.

I have some points that I would like to see corrected before acceptance.

Line 37 - What abnormalities? I think authors refer different fat taste perception/sensitivity in obesity, but as the sentence is constructed this is not totally clear. Please reformulate the sentence.

Lines 65 – 66 – “detection or identification threshold” - I suggest to omit this part. I do not find it essential at this point and I think it may cause a little confusion.

Line 86 – “lipids” - based on what is stated, I think it is more accurate to say “fatty acids” than “lipids”, at this stage, since are the fatty acids that bind the receptors in taste cells and evoke potentials.

Line 91 – “fat preferences” - Fat detection (at least fatty acid detection) is detected by receptors in tongue, but I think to say thar “fat preferences” are also brought by tongue receptors may be not accurate… Preference is a concept that involves more than just detection… I think it is not possible to argue that fat preference is due specifically to fatty taste detection at the level of receptors in taste cells, since it is not easy to separate the role of fatty acids detection at receptor level from mouthfeel and/or aroma perception…. If the authors intend to keep this statement of fat preference being dependent on the receptors at tongue level, please change the sentence to support this with studies showing this direct relation.

Lines 103-105 - I have some doubts about this relationship that authors made among different subjects… It is not because fat is stored as energy reserve that it means animals need lower detection/discrimination ability… When sugar is ingested in amounts higher than needed, to meet energy requirements, is also converted in fat to be stored… So fat stores do not only depend on fat ingestion… Even more, since this manuscript intends to be a review, even agreeing that it can be a critical review and authors can make hypothesis, I think it would be better to keep the hypothesis to the ones strongly supported by the studies that are being reviewed.

Line 128 – “to meet their energy needs” - From what has been said until now, probably better than “to meet their energy needs” can be “to acquire energy”? Because fat sensation (taste and mouthfeel) seems to result in intake even above the homeostatic energy needs, as has been discussed.

Line 158 - If not, at detection threshold, at what level? Recognition threshold? suprathreshold? hedonics? Please specify what type of perception is that that these authors found to be decreased in obesity.

Lines 177-178 - This does not mean a relationship with fat taste sensitivity. Please make the bridge between this sentence and what you have been stated. I suppose you mean that these results also support that obese people may have lower sensitivity to fat, since a high sensitivity may mean lower levels of fat to be preferred and lower sensitivity may require higher ingestion to the same reward feeling…

The next paragraph says that, but for a clear reading, this sentence needs to be better re-arranged with the rest of the text.

Line 180 – “suggested” and not “established”. As the authors reported before, there are no consensus among all studies done so far.

Line 184 - In line with my previous comment, this conclusion is to strong, taking into account what can be really concluded from all the studies. Moreover, I don’t think number 4 is a good note for this part… The authors use the sentence to make a summary of the section and, in that sentence, they push the reader to an example of hypothesis of how fat taste sensitivity and fat intake can work… I think this makes no sense at this point.

Section “high-fat diet and fat taste” - This section seems to be and extension of the previous section… I think the authors wanted to have a section linking the fat taste with obesity and another section linking the fat taste with fat consumption. However, when presenting the relationship between fat taste and obesity, they already referred  a link between fat intake and fat taste sensitivity… I agree that this can be necessary, since a part of the obesity can be related with fat intake. In that context, to omit thte influence of fat intake in the relationship between fat taste and obesity is not possible. Because of that, I suggest to change the order of the sections and to review/discuss before the relationship between fat intake and fat taste and only after the relationship between fat taste and obesity.

Lines 221-223 - This part of the sentence makes no sense in this section, since the authors are making this section (about the relationship between fat taste and fat consumption) separated from the other section (about the relationship between fat taste and obesity)

Section “fat taste and bariatric surgery” - This section must follow (or being included) the section about fat taste and obesity. This reinforces my comment of the need about re-structure the sequence of sections.

Line 250 - I consider the conclusions excessive (this happens in this summary and also in others). I would prefer “must be reversible”, since the authors stated that some studies fail in finding the same results, and some inter-individual variability exists.

Line 258 - please remove “this”

Lines 261-263 - This is repeating concepts that were already presented through the manuscript… Moreover, the title of this section indicates that studies showing that fat consumption may change reward system, and this sentence goes in line with reward system changing fat taste/consumption… Again, the text needs to be re-written

Lines 290-313 - please correct the format of the letters. They are different from the other format of the text

Author Response

Response to reviewers

Reviewer 1

We thank the reviewer for the useful comments and suggestions. In particular we have followed the recommendation to change the order of the different sections.

I have some points that I would like to see corrected before acceptance.

Line 37 - What abnormalities? I think authors refer different fat taste perception/sensitivity in obesity, but as the sentence is constructed this is not totally clear. Please reformulate the sentence. We have changed the sentence accordingly.

Lines 65 – 66 – “detection or identification threshold” - I suggest to omit this part. I do not find it essential at this point and I think it may cause a little confusion. The terms have been omitted as suggested.

Line 86 – “lipids” - based on what is stated, I think it is more accurate to say “fatty acids” than “lipids”, at this stage, since are the fatty acids that bind the receptors in taste cells and evoke potentials. This has been changed as suggested.

Line 91 – “fat preferences” - Fat detection (at least fatty acid detection) is detected by receptors in tongue, but I think to say thar “fat preferences” are also brought by tongue receptors may be not accurate… Preference is a concept that involves more than just detection… I think it is not possible to argue that fat preference is due specifically to fatty taste detection at the level of receptors in taste cells, since it is not easy to separate the role of fatty acids detection at receptor level from mouthfeel and/or aroma perception…. If the authors intend to keep this statement of fat preference being dependent on the receptors at tongue level, please change the sentence to support this with studies showing this direct relation. We agree with this remark and have deleted ‘fat preferences’ in the sentence.

Lines 103-105 - I have some doubts about this relationship that authors made among different subjects… It is not because fat is stored as energy reserve that it means animals need lower detection/discrimination ability… When sugar is ingested in amounts higher than needed, to meet energy requirements, is also converted in fat to be stored… So fat stores do not only depend on fat ingestion… Even more, since this manuscript intends to be a review, even agreeing that it can be a critical review and authors can make hypothesis, I think it would be better to keep the hypothesis to the ones strongly supported by the studies that are being reviewed. In agreement with the reviewer, this sentence has been deleted as not really related to the main message of the article

Line 128 – “to meet their energy needs” - From what has been said until now, probably better than “to meet their energy needs” can be “to acquire energy”? Because fat sensation (taste and mouthfeel) seems to result in intake even above the homeostatic energy needs, as has been discussed. We have followed the reviewer’s recommendation.

Line 158 - If not, at detection threshold, at what level? Recognition threshold? suprathreshold? hedonics? Please specify what type of perception is that that these authors found to be decreased in obesity. Details on this have been added.

Lines 177-178 - This does not mean a relationship with fat taste sensitivity. Please make the bridge between this sentence and what you have been stated. I suppose you mean that these results also support that obese people may have lower sensitivity to fat, since a high sensitivity may mean lower levels of fat to be preferred and lower sensitivity may require higher ingestion to the same reward feeling… We agree with this remark and have deleted the sentence (some references cited in this sentence have been added in the paragraph related to the reward system)..

The next paragraph says that, but for a clear reading, this sentence needs to be better re-arranged with the rest of the text. This paragraph has been rewritten.

Line 180 – “suggested” and not “established”. As the authors reported before, there are no consensus among all studies done so far. This has been changed.

Line 184 - In line with my previous comment, this conclusion is to strong, taking into account what can be really concluded from all the studies. Moreover, I don’t think number 4 is a good note for this part… The authors use the sentence to make a summary of the section and, in that sentence, they push the reader to an example of hypothesis of how fat taste sensitivity and fat intake can work… I think this makes no sense at this point. This has been changed.

Section “high-fat diet and fat taste” - This section seems to be and extension of the previous section… I think the authors wanted to have a section linking the fat taste with obesity and another section linking the fat taste with fat consumption. However, when presenting the relationship between fat taste and obesity, they already referred  a link between fat intake and fat taste sensitivity… I agree that this can be necessary, since a part of the obesity can be related with fat intake. In that context, to omit thte influence of fat intake in the relationship between fat taste and obesity is not possible. Because of that, I suggest to change the order of the sections and to review/discuss before the relationship between fat intake and fat taste and only after the relationship between fat taste and obesity. We thank the reviewer for these suggestions and have changed the order of the different sections in the new version of the manuscript.

Lines 221-223 - This part of the sentence makes no sense in this section, since the authors are making this section (about the relationship between fat taste and fat consumption) separated from the other section (about the relationship between fat taste and obesity). The title of the paragraph has been changed to make sense in its development. It now reads “Alteration of the reward system, high-fat diet and obesity”.

Section “fat taste and bariatric surgery” - This section must follow (or being included) the section about fat taste and obesity. This reinforces my comment of the need about re-structure the sequence of sections. Done has requested.

Line 250 - I consider the conclusions excessive (this happens in this summary and also in others). I would prefer “must be reversible”, since the authors stated that some studies fail in finding the same results, and some inter-individual variability exists. We have changed the sentence to take care of this comment and made the summary less strong.

Line 258 - please remove “this”. This has been removed

Lines 261-263 - This is repeating concepts that were already presented through the manuscript… Moreover, the title of this section indicates that studies showing that fat consumption may change reward system, and this sentence goes in line with reward system changing fat taste/consumption… Again, the text needs to be re-written. The paragraph has been rewritten.

Lines 290-313 - please correct the format of the letters. They are different from the other format of the text. This has been corrected.

Reviewer 2 Report

This review article by Drs. Brondel et al. is a very comprehensive review on Taste of fat and obesity. It is very well written and the material is presented in a logical manner.  The three leading ideas how the taste of fat is related to obesity are presented very well. However, the main issue with the review is that it is descriptive in nature and lacks a mechanistic component. This can be easily remedied by adding a chart or a diagram showing various mechanisms at the taste cell level and gut brain axis level that link the three leading ideas how the taste of fat is related to obesity.

Some of sentences in the text are very long and can be shortened to improve readability.

Author Response

Response to reviewers

Reviewer 2

This review article by Drs. Brondel et al. is a very comprehensive review on Taste of fat and obesity. It is very well written and the material is presented in a logical manner.  The three leading ideas how the taste of fat is related to obesity are presented very well. However, the main issue with the review is that it is descriptive in nature and lacks a mechanistic component. This can be easily remedied by adding a chart or a diagram showing various mechanisms at the taste cell level and gut brain axis level that link the three leading ideas how the taste of fat is related to obesity. A figure has been added

Some of sentences in the text are very long and can be shortened to improve readability. Done as requested.
